# γδ T Cells in Emerging Viral Infection: An Overview

**DOI:** 10.3390/v14061166

**Published:** 2022-05-27

**Authors:** Eleonora Cimini, Chiara Agrati

**Affiliations:** Laboratory of Cellular Immunology and Pharmacology, National Institute for Infectious Diseases, “L. Spallanzani”, Via Portuense 292, 00149 Rome, Italy; eleonora.cimini@inmi.it

**Keywords:** gammadelta T cells, innate immunity, antiviral activity, coronaviruses, flaviviruses, filoviruses

## Abstract

New emerging viruses belonging to the *Coronaviridae*, *Flaviviridae*, and *Filoviridae* families are serious threats to public health and represent a global concern. The surveillance to monitor the emergence of new viruses and their transmission is an important target for public health authorities. Severe acute respiratory syndrome coronavirus-2 (SARS-CoV-2) is an excellent example of a pathogen able to cause a pandemic. In a few months, SARS-CoV-2 has spread globally from China, and it has become a world health problem. Gammadelta (γδ) T cell are sentinels of innate immunity and are able to protect the host from viral infections. They enrich many tissues, such as the skin, intestines, and lungs where they can sense and fight the microbes, thus contributing to the protective immune response. γδ T cells perform their direct antiviral activity by cytolytic and non-cytolytic mechanisms against a wide range of viruses, and they are able to orchestrate the cellular interplay between innate and acquired immunity. For their pleiotropic features, γδ T cells have been proposed as a target for immunotherapies in both cancer and viral infections. In this review, we analyzed the role of γδ T cells in emerging viral infections to define the profile of the response and to better depict their role in the host protection.

## 1. Introduction

New emerging viruses, such as Ebola, Zika, and Dengue viruses, severe acute respiratory syndrome (SARS) coronavirus, Middle East respiratory syndrome (MERS) coronavirus, the Avian influenza virus, and the latest severe acute respiratory syndrome coronavirus-2 (SARS-CoV-2), responsible for the Coronavirus disease-19 (COVID-19) pandemic, are serious threats to public health and have become a global concern. Changes in the environment have led to an increase in contact between wild animals and humans. Surveillance of new viruses’ emergence and their transmission are an important target for public health authorities. Despite extraordinary advances in the development of countermeasures (diagnostics, therapeutics and vaccines), the increase in global interdependence and in world travel have added complexity to containing these infectious diseases, which could affect not only the health but also the economy of societies [1]. SARS-CoV-2 is an excellent example of this condition becoming a global health problem in a few months, and all countries are helping contain this pandemic, which has been going on for two years now.

γδ T cells are characteristic lymphocytes, with both innate and adaptive immune qualities [2,3]. In humans, γδ T lymphocytes constitute a small population found in circulation and tissues (<10%) [4]. Generally, γδ T cells are divided into Vδ2 and non-Vδ2 γδ T cells populations, and Vδ1 represent the majority of non-Vδ2 T cells, according to their TCR expression [5,6,7]. Vδ1 T cells are resident lymphocytes in mucosal surfaces and epithelia of the intestine, of lungs, and of the urogenital tracts; on the other hand, Vδ2 T cells represent 90% of the total circulating γδ lymphocytes in peripheral blood of healthy donors [8]. γδ T cells are able to recognize antigens not necessarily depending on MHC antigen presentation [9,10]. In humans, Vδ1+ T cells can react with CD1 molecules, the MHC-related protein 1 (MR-1), the endothelial protein C receptor (EPCR), the UL16-binding protein 4 (ULBP4), and MHC class I chain-related protein A and B (MICA and MICB), induced on stressed intestinal epithelial cells [10,11]. Human Vγ9Vδ2 T cells recognize phosphoantigens (pyrophosphate-containing organic molecules, PhAg), such as (E)-4-hydroxy-3-methyl-but-2-enyl pyrophosphate (HMBPP), present in many pathogens, and isopentenyl pyrophosphate (IPP), which accumulates in tumors or cells treated with aminobisphosphonates, such as Zoledronic Acid [12]. PhAg sensing and activation of TCR is mediated by a subclass of Butyrophilin (BTN)-related molecules (BTN3A1-3); some of them can control γδ T cell subset homeostasis and activation and can bind their TCR [13]. In particular, BTN2A1 and BTN3A1 are necessary for Vγ9Vδ2 T cell activation via phosphoantigen recognition [14,15]. Both Vδ1 and Vδ2 T cells present a potent antiviral activity [16,17] and anti-tumor properties [18,19]. Indeed, γδ T cells react to viral entry quickly, but the precise mechanism deployed by human γδ T cells in response to viral infections is still not clear. Their ability in early sensing of the infection, the quick activation, and cytotoxicity against different types of viruses, such as influenza A virus, hepatitis B (HBV) and C (HCV) virus, cytomegalovirus (CMV), human immunodeficiency virus (HIV), and severe acute respiratory syndrome-related coronavirus (SARS-CoV), make them suitable as a good target for therapies [19,20]. According to their specific features (such as their non-MHC restricted recognition, the quick production of abundant cytokines, and potent cytotoxicity in response to different malignancies), γδ T-cell-based cancer immunotherapy has great promise in tumor therapy [21,22].

In this review, we will analyze the role of γδ T lymphocytes toward emerging viruses to better depict their antiviral activity in host protection.

## 2. Coronaviruses

Human coronaviruses (HCoVs) were first discovered in the 1960s, named HCoV-OC43 and HCoV-229E, and in 2004 and 2005, named HCoV-NL63 and HCoV-HKU1, respectively. HCoVs are generally considered relatively harmless viruses. The past two decades have seen the emergence of three zoonotic CoVs, which have jumped species to cause lethal diseases in humans: SARS-CoV, MERS-CoV, and SARS-CoV-2, responsible for the COVID-19 pandemic. These three viruses belong to the *Coronaviridae* family and are large, enveloped, single-stranded RNA viruses. They are the largest known RNA viruses (118–136 nm virion diameter). Virions are spherical and have a large spike (S) glycoprotein that extends 16–21 nm from the virus envelope. The family is divided into two subfamilies, the *Torovirinae* and the *Coronavirinae* [23], whose members are widespread among mammals, causing often mild respiratory or enteric infections. Over 60 CoVs were isolated from bats (BtCoV) belonging to genus betacoronavirus. Bats are large CoV reservoirs, and many bat species have their own unique BtCoV, suggesting a long history of co-evolution. An involvement of human γδ T cells was known in SARS-CoV and SARS-CoV-2 infections, but there are no data in the literature regarding MERS-CoV.

### 2.1. Severe Acute Respiratory Syndrome Coronavirus (SARS-CoV)

A viral outbreak, which occurred in China in 2002, was linked to an infection with a new CoV (SARS-CoV) [24]. In September 2003, SARS-CoV infected more than 8000 people in 29 countries with 774 deaths around the globe; about 30% of the infected people were severe cases, and 20% were healthcare workers (HCWs) [25]. SARS-CoV is transmitted from its natural reservoir in bats through several animal species (e.g., civet cats, raccoon dogs sold for human consumption in markets in southern China) [26]. The main transmission route is by close person-to-person contact, through respiratory droplets. Many people may develop pneumonia, and, in some cases, SARS can be fatal because of respiratory failure [27].

Healthcare workers are a high-risk group, and a strong association between advanced age and disease severity was observed [28]. However, the presence of long-lived neutralizing antibodies and memory T and B cells in convalescent SARS-CoV patients represented a hope for active immunization. Inflammatory responses are characterized by upregulation of proinflammatory cytokines/chemokines in tissues and serum (IL-6, IP-10, and MCP-1) and by a massive infiltration of the macrophages in infected tissues [29].

Regarding T-cell response against SARS-CoV, most SARS-specific CD4 cells showed a central memory phenotype and produced mainly one cytokine. SARS-specific CD8 responses are IFN-γ and TNF-α positive cells, able to degranulate cytotoxic soluble factors [30]. After three months from the disease onset, the healthcare workers who experienced SARS-CoV infection had a selective expansion of circulating Vδ2 γδ T cells [31] able to kill in vitro SARS-CoV infected target cells in an IFN-γ-dependent way. Their increase was proportional with anti-SARS-CoV IgG titers, suggesting their protective role during coronavirus infections [31]. The antiviral activity of Vδ2 T cells from SARS-CoV-1-infected patients needs further confirmation.

### 2.2. Severe Acute Respiratory Syndrome Coronavirus-2 (SARS-CoV-2)

Coronavirus disease 2019 (COVID-19) is a highly infectious type of pneumonia caused by SARS-CoV-2, which spread throughout China and, subsequently, across the world, rapidly becoming a global pandemic since December 2019. As of 8 April 2022, 494,587,638 cases are confirmed, together with 6,170,283 deaths globally since the beginning of the COVID-19 pandemic. As of 5 April 2022, a total of 11,250,782,214 vaccine doses have been administered [WHO Coronavirus (COVID-19) Dashboard]. COVID-19 has been a serious public health concern across the world and demanded urgency in basic science, clinical research, and vaccine strategies. Regarding the genetic homology, SARS-CoV-2 is close to SARS-CoV (79% nucleotide identity) and also to MERS-CoV (51.8% nucleotide identity) [32]. The pathophysiology of SARS-CoV-2 infection is similar to SARS-CoV, i.e., signs of atypical pneumonia with progression to acute respiratory distress syndrome (ARDS) in some individuals were observed and were characterized by high inflammatory responses in the lower airways responsible for 28% of fatal COVID-19 cases. The SARS-CoV-2 infection and the dysregulation of immune response are characteristic of the severe COVID-19 with a panel of multi-organ damages (cardiac, renal and hepatic systems) [33].

Innate immunity is able to clear the virus in the upper airways of COVID-19 patients in the first 10–12 days from infection, inducing the resolution of the infection [34] in the majority of individuals [35]. Innate immune cells (macrophages, monocytes, dendritic cells, neutrophils, NK cells, and γδ T cells) expressed several PRRs that recognize PAMPs or damage-associated molecular patterns (DAMPs) and trigger inflammatory signaling pathways and immune responses to fight the virus [36]. In COVID-19 patients, we observed an increased production of inflammatory cytokines paralleled with an increase in myeloid-derived suppressor cells (MDSC). These frequencies inversely correlated with perforin-expressing NK and T cells. Indeed, the frequency of NK cells positive for perforin in intensive care unit (ICU) patients was lower when compared with non-ICU patients, suggesting a strong impairment of the immune cytotoxic arm, at least partially mediated by MDSC expansion [37,38].

γδ T cell frequency in hospitalized COVID-19 patients is lower than in healthy controls [39], according to the lymphocytopenia (frequency of lymphocytes) observed in severe COVID-19 patients. Interestingly, a differentiation toward the effector (memory) γδ T cells more capable of tissue infiltration has been observed 2 weeks after hospital admission [40]. The reduction in circulating γδ T cells has been, therefore, associated with their recruitment in the airway tissues [41]. Accordingly, Saris A. et al. showed a high frequency of γδ T cells in BALF of COVID-19 patients admitted to ICU in comparison to the circulating compartment, and their frequency increased with the persistence of clinical severity [42]. Moreover, circulating and resident-tissue γδ T cells showed an activated phenotype at the injury epicenter in COVID-19 patients with respect to healthy controls [43]. The upregulation of the activation/exhaustion markers CD25 [44] and CD69 and PD-1 on γδ T cells in COVID-19 patients was observed [45,46]. The involvement of γδ T cells in COVID-19 is further supported by an increase in IL-18 [46], a cytokine that seems to be involved in γδ T-cell activation in viral infections [47]. Furthermore, a paper from Flament et al. demonstrated that MAIT cell phenotype correlated with innate lymphoid cells and γδ T-cell alteration in patients with COVID-19 [48].

Another study showed the immune signatures of 63 COVID-19 patients and 55 healthy controls, highlighting that γδ T-cell depletion in the blood of patients was proportional to the disease severity [49], suggesting that it could be used as a diagnostic/prognostic marker. Similar approach has also been reported by another study, showing that a Neutrophil/Vδ2 ratio is a better prognostic marker of COVID-19 severity than the Neutrophil/CD8+ Lymphocytes ratio [50]. Interestingly, in COVID-19 patients, an expansion of a CD16+ γδ T-cell population was observed in a single-cell transcriptional profiling and was associated with moderate disease [51].

No data are available about the specific role of γδ T cell during an Omicron viral infection.

γδ T cells seem to be involved in children with COVID-19 [52], and they are a predominant immune response in early life [53]. A strong γδ T-cell activation was observed in a pediatric setting [54], indicating a strong involvement of γδ T cells in the immune response to COVID-19 in children. In a recent paper, the authors demonstrated, using a scRNAseq assay, the differentiation profile of γδ T cells in the lungs of cancer patients and in COVID-19 patients. Interestingly, the authors demonstrated that γδ T cells (Vγ9 and non-Vγ9) of severe COVID-19 adult patients are recruited in the lesions of the lung from peripheral blood and showed an effector/exhausted phenotype. In addition, in pediatric acute COVID-19 patients, the authors observed a γδ T-cell lymphopenia similar to the adult patients [55].

All these data contribute to describing a key function of γδ T cells against SARS-CoV-2 infection, even if other studies are necessary to better depict γδ T-cell role against this emerging pathogen.

## 3. Flavivirus

*Flaviviridae* is a large family of unsegmented positive-strand RNA viruses and contains three genera: Flavivirus, Hepacivirus, and Pestivirus. Members of the Flavivirus genus cause widespread morbidity and mortality throughout the world. Some of the mosquitoes-transmitted viruses include: Yellow Fever, Dengue Fever, West Nile virus, Japanese encephalitis, and Zika virus. Other Flaviviruses are transmitted by ticks: Tick-borne Encephalitis (TBE), Kyasanur Forest Disease (KFD), Alkhurma disease, and Omsk hemorrhagic fever, responsible for encephalitis and hemorrhagic diseases [56]. Virions of Flavivirus are enveloped and spherical (diameter 40–60 nm). Well-characterized species of this family are the Classical Swine fever virus, the Yellow fever virus, and the human HCV [57]. All members of the genus Flavivirus are closely related and share significant aminoacid sequence identity, which results in serological and cellular response cross-reactivity [58]. Depending on the antigenic properties of the viruses, when a person is exposed, the flavivirus cross-reactivity can be beneficial or could promote immune pathologies [59]. The presence of reactive CD8 and CD4 T cells specific for different viral epitopes of Flavivirus is well known [60]. On the other hand, the innate immune response to Flavivirus has been structured by the IFN system, plasmacitoid dendritic cells, Langherans Cells, Natural killer cells, mast cells, monocytes/macrophages [61,62], and γδ T cells [16]. We demonstrated an effective γδ T-cell response in HCV-infected patients and in an in vitro model of HCV-infected liver culture [63,64,65]. The ability of γδ T cells to inhibit other members of the *Flaviviridae* family is described below.

### 3.1. Dengue Viruses

Dengue virus (DENV) has become one of the major emerging infectious diseases worldwide, resulting in around 100–400 million cases every year [66]. It is a rapidly spreading vector-borne disease with high mortality rates. The infection causes acute febrile illness, a major public health concern in the tropics and subtropics areas. The mosquito vector is called Aedes aegypti and transfers the virus to humans. The Flavivirus genus consists of four different types of DENV viruses, starting from DENV-1 to DENV-4 [67], and it is found in urban and semiurban areas of tropical and subtropical bands [68,69]. The incubation period is 3 to 14 days, while the period of illness is 3 to 7 days [70]. Dengue fever symptoms include nausea, vomiting, and fever. Severe Dengue disease is characterized by severe headache, retro-orbital and muscle pain, joint and bone pain, macular or maculopapular rash, and minor hemorrhagic manifestations [71].

In primary DENV infection, both the innate and the acquired responses collaborate to clear the virus, but some of these responses may in turn cause host damage, through an event known as the “cytokine storm” characterized by an imbalance between the Th1 and Th2 cytokine responses [72].

Human γδ T cells have a protective role during DENV infection by establishing an immunological interplay with mast cells to kill DENV-infected cells by producing IL-18 [47]. Furthermore, γδ T cells were able to kill DENV-infected targeted DCs and contributed to virus clearance in vivo in a mouse model [73]. In DENV-infected patients, Vδ2 T cells were reduced and expressed high levels of CD38 and HLA-DR [74] activation markers, similarly to NK cells in the first phase of infection [75]. In DENV-infected patients, we demonstrated that IFN-γ production by γδ T cells was impaired after phosphoantigen stimulation, but they were enriched in perforin. Furthermore, the reduced cytokine release capability may depend on TIM-3 expression, as we demonstrated a negative correlation between the expression of TIM-3 and IFN-γ production by PhAg-stimulated Vδ2 T cells from DENV patients [74].

### 3.2. Zika Virus

Zika virus is a mosquito-borne pathogen and an enveloped, icosahedral, positive-sense, single-stranded non-segmented RNA virus belonging to the Flavivirus genus. It is spread by Aedes mosquitoes (*A. aegypti* and *A. albopictus*). Its name comes from the Ziika (Zika) Forest in Uganda, where the virus was first isolated in 1947 [76]. From 2007 to 2016, the virus spread across the Pacific Ocean to the Americas, causing a large epidemic in 2015–2016 [77]. The infection, known as Zika fever or Zika virus disease, often causes no or only mild symptoms, similar to a very mild form of dengue fever. Symptoms include fever, joint pain, headache, red eyes, and a maculopapular rash. Infection in adults has been linked to the Guillain–Barré syndrome (GBS), and some of the complications include encephalitis, meningoencephalitis, meningitis, myelitis, and neuropsychiatric symptoms [78,79,80,81]. A pregnant woman can infect her baby with Zika, resulting in microcephaly and severe brain malformations [82,83,84,85]. Zika can also be transmitted from men and women to their sexual partners [86,87], and studies of screening of blood donors demonstrated the detection of ZIKV in blood of asymptomatic French Polynesia donors [88]. This epidemic was marked mainly by the exponential growth in microcephaly cases and other congenital defects, resulting in an epidemic of public health importance due to the pre-existing immunity of other flaviviruses and the potential immunopathogenesis [89].

The innate immune response is the first line of defense, and it recognizes pathogens by PRR receptors triggering macrophages, dendritic cells, natural killer cells, and endothelial cells to produce several mediators, which modulate viral replication and immune response to the virus [90]. During ZIKV infection, CD4+ T cells differentiate into Th1 cells able to produce IFN-γ, IL-2, TNF-α cytokines, and transcription factor T-bet. Effector CD8+ T cells also produced IFN-γ and TNF-α, with increased expression of granzyme B on them [91]. The cellular innate response is mediated by dendritic cells, macrophages, NK cells, and γδ T cells [92,93]. Regarding γδ T cells in acute ZIKV infection, a substantial expansion of CD3+CD4-CD8-T-cell subset expressing Vδ2 TCR was observed. Vδ2 T cells were characterized by a terminally differentiated profile, enriched in granzyme B and able to produce IFN-γ after PhAg stimulation [94]. According to these data, we defined in vitro the antiviral role of Vδ2 T cells against ZIKV. We demonstrated that ZIKV infection induced Vδ2 T cell expansion and sensitized tumor cells to Vδ2-mediated killing. Furthermore, phosphoantigen-expanded Vδ2 T cells were able to kill ZIKV-infected cells with perforin release by the NKG2D/NKG2DL pathway [95]. These data showed a strong antiviral activity of Vδ2 T cells against ZIKV-infected cells, suggesting their involvement in the protective immune response. To study whether the role of the expanded Vδ2 T cells in vivo could be associated with disease complications, other studies are mandatory.

### 3.3. West Nile Virus (WNV)

West Nile virus (WNV) is an enveloped and positive single-stranded RNA belonging to the Flavivirus genus. WNV is endemic in parts of Africa, Europe, the Middle East, and Asia, and since 1999, it has spread to North America, Mexico, South America, and the Caribbean [96]. Peak transmission occurs during summer months. The virus is transmitted to humans and animals by Culex mosquitoes. In general, the transmission cycle is maintained between mosquitoes and birds, and occasionally, the virus can spread to horses and humans, resulting in serious disease and even death [97]. The majority of the cases are asymptomatic. Symptoms (fever, headaches, nausea, vomiting, muscle aches, including a rash and swollen glands) usually last a few days to several weeks. Severe disease includes high fever, disorientation, tremors, convulsions, paralysis, and coma that can cause neurological damage. In rare cases, the illness can be fatal.

The elderly and immunocompromised individuals are at a higher risk of developing the West Nile neuroinvasive disease [98]. The role of γδ T cells in the protective immune response against WNV was demonstrated in in vivo studies of WNV meningoencephalitis in laboratory mice [99]. In humans, we demonstrated a strong antiviral activity of bisphosponate-activated Vδ2 T cells, able to produce a large amount of IFN-γ antiviral soluble factor. Furthermore, Vδ2 T cells can efficiently kill WNV-infected cells through perforin release. Altogether, these results provide insight into the effector functions of human Vδ2 T cells against WNV [99].

## 4. Filovirus

Filoviruses are non-segmented negative-stranded RNA viruses belonging to the order Mononegavirales and are different from other viruses in terms of genetic, morphological, and biological characteristics [100]. At present, members of the *Filoviridae* family are classified into five genera: Ebola virus (EBOV), Marburg virus (MARV), Stria virus, Thamno virus, and Cueva virus, with the proposal for a sixth genus, Dianlo virus [101]. The Ebolavirus genus consists of six virus species (Zaire ebolavirus, Sudan ebolavirus, Tai Forest ebolavirus, Reston ebolavirus, Bundibugyo ebolavirus, and Bombali ebolavirus). EBOV and MARV are considered the most virulent members of the *Filoviridae* family among the total 12 filoviruses genera. EBOV and MARV have caused several human and animal outbreaks across the world [102] and are listed as high-priority pathogens by the WHO because of their epidemic potential [103]. Filovirus disease outbreaks have caused significant loss of human and animal lives. The largest filovirus disease outbreak occurred in 2013–2016 during the Ebola virus disease (EVD) outbreak, which began in west Africa and later spread to other parts of the world [104]. In total, 28,652 cases with 11,325 deaths in 10 countries occurred, with 99% of the fatalities occurring in neighboring Guinea, Sierra Leone, and Liberia [105]. The 2013–2016 outbreak was classified by the WHO as a Public Health Emergency of International Concern, by focalizing the questions on the management of these infections with ebolaviruses to better prepare the surveillance in future epidemics. Data on the immune response to EVD were very difficult to obtain due to the limitations imposed by biosafety requirements and logistics.

We analyzed the innate and specific response in peripheral blood mononuclear cells of the two Italian Ebola-infected patients admitted to the National Institute for Infectious Diseases “L. Spallanzani” in Rome. We investigated the kinetics and the functionality of T-cell subsets during the acute phase of EVD until recovery by multiparametric flow cytometry. A sustained decrease in CD4 T cells in both patients and a high T-cell activation were observed, associated with autophagic/apoptotic phenotype, enhanced expression of PD-1 (exhaustion marker), and impaired production of IFN-γ [106], according to the immune signature identified in EVD fatalities and survivors in Guinea [107].

Finally, following the previous study on T-cell response in EVD patients, we investigated γδ T-cell involvement in vivo in EVD patients from admission in the Ebola Treatment Center in Guinea until their discharge/death to assess its association with the clinical outcome. Results showed a low frequency of Vδ2 T cells during EVD, independently of the clinical outcome. Moreover, Vδ2 T cells from EVD patients expressed a high percentage of CD95 apoptotic marker. Furthermore, Vδ2 T cells from survivors were effector cells with a lower expression of CTLA-4 exhaustion marker than the fatalities, suggesting a role of Vδ2 T cells in host protection [108]. Our results showed that effector Vδ2 T cells were associated with survival, suggesting their involvement in the complex network of protective response to EBOV infection.

## 5. Conclusions

Emerging viruses represent a global and social concern. This review focused on summarizing the role of γδ T cells in fighting several emerging infections with the aim to define similar patterns of response. In recent years, several papers focused on γδ T-cell involvement during single emerging viral infection. Here, we summarized the existing data on γδ T-cell response during the past and recent emerging virus epidemics/pandemics by highlighting the γδ T cells’ antiviral role as “innate immune players” with multiple peculiarities (Figure 1). They quickly sense several emerging pathogens and contribute to the host’s protection by exerting both cytolytic and non-cytolytic strategies and by shaping other immune cells to fight these viruses. Moreover, γδ T cells are not viral specific cells, but they recognize host molecules induced by cancers and infections, making them able to quickly respond to old and emerging infections. Their immune peculiarities make them feasible to be a target for future cellular immunotherapies in viral infections.

## Figures and Tables

**Figure 1 viruses-14-01166-f001:**
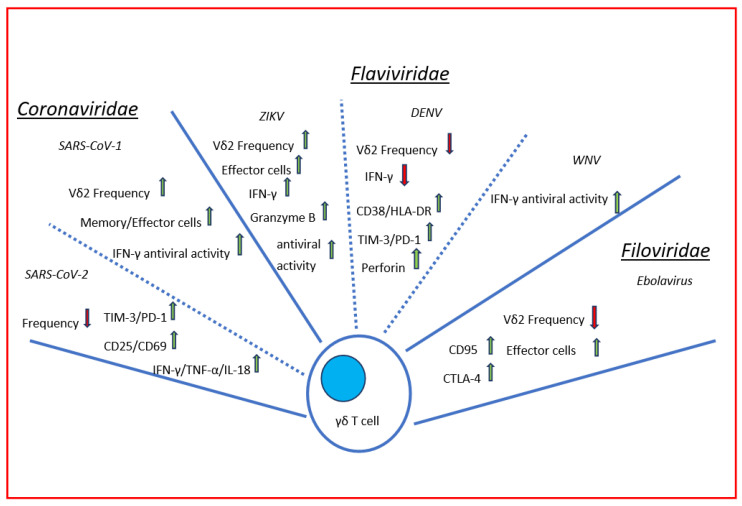
Immune response of human γδ T cells to viral emerging infections. Briefly, γδ T cells are able to recognize cells infected with emerging viruses belonging to *Coronaviridae*, *Flaviviridae,* and *Filoviridae* families and exert their antiviral activity by cytolytic and non-cytolytic mechanisms. During the infection, γδ T-cell frequency decreased/increased depending on the emerging virus. Regardless, γδ T cells differentiate in activated memory/effector (CD25, CD69, CD38, HLA-DR, and CD95 expression) cells after interaction with the pathogen and release large amount of pro-inflammatory cytokines (IFN-γ, TNF-α, IL-18) and cytotoxic soluble factors (Perforin and Granzyme B). Through all these factors, γδ T cells exert their potent antiviral activity. During the infections, γδ T cells express exhaustion markers (PD-1, TIM-3), as other circulating T αβ cells, dampening their effector functions.

## Data Availability

Not applicable.

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
