# Peer review of "γδ T Cells in Emerging Viral Infection: An Overview"

_viruses, 2022, doi:10.3390/v14061166_

Round 1

Reviewer 1 Report

This aim of this review paper is an update of the works on the status of gamma delta T cells in the specific context of emerging viral infections. In its present form the paper needs important corrections. Some references are either inaccurate or missing. Some statements require moderation. Many sentences are unclear and mispellings make it sometimes difficult to read.

Majors corrections :

Introduction

L.41-45: Vd1 should be replaced by non-Vd2; recent reviews should be given for references rather than isolated non-exhausting original works (refs 5, 6).

L.45-46: as it is, the statement is inaccurate as many gd cells do not recognize soluble antigens but surface molecules(Deseke2020).

L.46-48: many Vd1 cells were shown to recognize molecules different from MICA/B (Deseke2020)

L48-53: update information and incorporate the implication of BTN2A1. Suggestion: add a sentence on the role of BTNs in general in gd T cell biology.

L52-53: cited references do not deal with PhAg sensing…Others should be chosen.

L54: in place of or in addition to reference 12 (2005), a more recent review could be cited.

Section 2.1 : statements in lines 86-89 and 90-95 have no references.

Lines 100-103: the statement that Vd2 cells from SARS-CoV patients are able to kill infected targets, derived from ref.12, is misleading as the paper showed that Vd2 cells from healthy patients were able to do so ; moreover they were doing this after in vitro activation with Zoledronate. Thus there was no demonstration of their direct activity in vivo and this should be discussed.

Section 2.2:

L.121: does “innate immunity” include gd cells in this statements which cites ref.24 (cf. lines 224-225)? The mechanisms involved should be explicated and it would be important in this section to quote and discuss the original data in addition to quoting a review.

L.151-152:  ref41 is unrelated to CoViD infection. Thus this interesting reference should be cited or discussed elsewhere.

The paper by Cerapio et al (Viruses, 2021) is not presented (DOI: 10.3390/v13112212)

The paper by Flament et al, Nat Immunol 2021 ( DOI: 10.1038/s41590-021-00870-z) also reports gd cell alteration in COVID19, (in addition to MAIT cells) an is not presented.

Section 3:

Since HCV is a flavivirus and there is significant literature about gd and HCV (in particular from the author’s group), it would be of interest to question the differences and similarities between gd cell responses in HCV and other flaviviruses.

L.195-197: alterations concerning gd frequency and activation markers is unclear; it is not said how frequencies are altered.

L.199-200: the direct link between TIM-3 expression and exhaustion should be discussed or presented more cautiously.

L.219: it should be explained how “Innate immunity” (which component ?) is important for ZIKV “pathogenesis”. Do the authors mean pathophysiology rather than pathogenesis ? or protection against ZIKV?

L251-252:  it should be precised if studies in mice used “human” gd T cells and how this can be extrapolated to humans since the similarities between human and murine gd cells are subject of debate.

L.256: ref 84 deals with EBOV, not WNV.

Figure:

It should be explicited what “Zol-mediated” refers to (in the case of WNV). If this means that zol-activated Vd2 cells can kill WNV-infected cells, the situation is the same for SARS-coV-1. In the case of ZIKV, phosphoantigen-activated cells are said to have similar activity, thus this could be indicated as well (cf lines 231-232).

 Minor corrections

General comment (suggestion):

It might be interesting to discuss how the study of these “emerging viruses” adds to the current knowledge of the role of gd T cells in viral infections and how responses in emergent infections differ from responses in “old viruses“ infections ?

Additional spelling or syntax errors (among many others), or sentences benefiting reformulation:

L.41 :  circulating BLOOD ?

L.42: According to ?

L62 : “these emerging viruses” : seems to refer to the ones listed at the beginning of introduction and not to the ones just listed before and comprising HIV, HBV, HCV, IAV. This must be clarified.

L.72: Virions are spherical and HAVE a large spike.

L.79: there are NO data in THE literature concerning their involvement against MERS-CoV.

L.82:  In September 2003, the SARS-CoV had infected = remove pandemic

L86-89   Keep the transmission route in one sentence and symptoms in a separate sentence

L.99: Health care workers WHO experienced SARS-CoV

L.107: On April 8th 2022, 494.587.638 cases are confirmed together with 6.170.283 deaths

L.19: As of April 5th 2022

L.110: COVID-19 is A serious public health concern AND demanded

L.117: response, remove “and”

L.119: damages

L.130: γδ.T cell count  …?  L.133: 2 weeks after hospital admission

L.135: a high frequency.. in comparison to circulating

L.138: phenotype

L.153: ref 43 did not look at γδ  distribution/Activation in regard to COVID-19 infection

L.170: of the virus to which ?

L.172:  the sentence is either incomplete or improperly phrased.

L.185: “Backbone fever” is not appropriate as a symptom.

L.190: cytokine storm resulting in ?

L.198: VD2 T cells IFN-g production is impaired?

L.203: belonging to

L.204: Zika forest

L.212: transfer ? Zika to the baby resulting in…

L.214: and studies

L.214: in blood of asymptomatic French Polynesia donors

L.220: which encode

L.222: T cells differentiate into Th1

L.228: enriched in granzyme B and able to produce

L.244: humans resulting in

L.246: Severe disease includes

L.249: Do the authors mean that WNV infects the CNS and causes severe disease in immunocompromised or elderly patients ? The sentence is not clear.

L.250: A role for γδ  t cells in

L.272: with 99% of fatalities occurring in neighbouring …

L.288: discarge ???

L.290: independently of the clinical outcome

L.297: belonging to

Author Response

Reviewer 1:

Majors corrections:

Introduction

Question 1: L.41-45: Vd1 should be replaced by non-Vd2; recent reviews should be given for references rather than isolated non-exhausting original works (refs 5, 6).

Answer 1: According to the reviewer’s suggestion, we modified the sentence in the Introduction section lines 41-43 we added 2 new ref in the Reference list n 6, 7

Question 2: L.45-46: as it is, the statement is inaccurate as many gd cells do not recognize soluble antigens but surface molecules (Deseke 2020).

Answer 2: According to the reviewer’s suggestion, we modified the sentence in the Introduction section lines 46-47 we added 2 new ref in Reference list n 9, 10

Question 3: L.46-48: many Vd1 cells were shown to recognize molecules different from MICA/B (Deseke2020)

Answer 3: We modified the test accordingly, lines 47-50

Question 4: L48-53: update information and incorporate the implication of BTN2A1. Suggestion: add a sentence on the role of BTNs in general in gd T cell biology.

Answer 4: we added a sentence on the role of BTNs in gd T cell biology lines 56-58 and in Reference list 2 new ref n. 14-15

Question 5: L52-53: cited references do not deal with PhAg sensing…Others should be chosen.

Answer 5: we changed le references accordingly to the reviewer’s suggestion. Line 56-58

Question 6: L54: in place of or in addition to reference 12 (2005), a more recent review could be cited.

Answer 6: According to the reviewer’s suggestion, we added a recent review (n 17)

Section 2.1:

Question 1: statements in lines 86-89 and 90-95 have no references.

Answer 1: According to the reviewer’s suggestion, we added the references numbers 27-29

Question 2: Lines 100-103: the statement that Vd2 cells from SARS-CoV patients are able to kill infected targets, derived from ref.12, is misleading as the paper showed that Vd2 cells from healthy patients were able to do so; moreover they were doing this after in vitro activation with Zoledronate. Thus there was no demonstration of their direct activity in vivo and this should be discussed.

Answer 2: in the paper (ref 21. Poccia, F.; Agrati, C.; Castilletti, C.; Bordi, L.; Gioia, C.; Horejsh, D.; Ippolito, G., Chan, P.K.S.; Hui, D.S.C.; Sung, J.J.Y. et al. Anti-severe acute respiratory syndrome coronavirus immune responses: the role played by V gamma 9V delta 2 T cells. J Infect Dis 2006, 193:1244-9. doi: 10.1086/502975), the authors demonstrated that SARS-CoV-1 infection in HCWs induced an expansion of Vd2 T cells expressing an effector phenotype. Moreover, they demonstrated that ex vivo gd T cells or gd T cells lines obtained from PBMC of healthy controls can kill SARS-CoV-1-infected cells. We added a sentence at lines 113-114.

Section 2.2:

Question 1: L.121: does “innate immunity” include gd cells in this statements which cites ref.24 (cf. lines 224-225)? The mechanisms involved should be explicated and it would be important in this section to quote and discuss the original data in addition to quoting a review.

Answer 1: We added a sentence to clarify cellular innate immune response (lines 134-138).

Question 2: L.151-152:  ref41 is unrelated to CoViD infection. Thus this interesting reference should be cited or discussed elsewhere.

Answer 2: We agree with the reviewer and corrected the ref accordingly

Question 3: The paper by Cerapio et al (Viruses, 2021) is not presented (DOI: 10.3390/v13112212)

The paper by Flament et al, Nat Immunol 2021 ( DOI: 10.1038/s41590-021-00870-z) also reports gd cell alteration in COVID19, (in addition to MAIT cells) an is not presented.

Answer 3: According to the reviewer’s suggestion, we added these refs with two comments, lines 157-159 and 172-178 and the related References

Section 3:

Question 1: Since HCV is a flavivirus and there is significant literature about gd and HCV (in particular from the author’s group), it would be of interest to question the differences and similarities between gd cell responses in HCV and other flaviviruses.

Answer 1: According to the reviewer’s suggestion, we added a sentence (line 199-202) e 3 new ref (n 63-65).

Question 2: L.195-197: alterations concerning gd frequency and activation markers is unclear; it is not said how frequencies are altered.

Answer 2: According to the reviewer’s suggestion, we modified the text (line 223)

Question 3: L.199-200: the direct link between TIM-3 expression and exhaustion should be discussed or presented more cautiously.

Answer 3: According to the reviewer’s suggestion, we added a sentence, line 227-229.

Question 4:L.219: it should be explained how “Innate immunity” (which component ?) is important for ZIKV “pathogenesis”. Do the authors mean pathophysiology rather than pathogenesis ? or protection against ZIKV?

Answer 4: we modified the sentence according to the reviewer’s suggestion line 248-251 and we added a new ref in reference list (n90).

Question 5: L251-252:  it should be precised if studies in mice used “human” gd T cells and how this can be extrapolated to humans since the similarities between human and murine gd cells are subject of debate.

Answer 5:  In the studies about the in vivo activity of gd T cell in mice, the author did not used humanized gd T cells.

Question 6: L.256: ref 84 deals with EBOV, not WNV.

Answer 6: we corrected accordingly, line 281

Figure:

Question 1: It should be explicited what “Zol-mediated” refers to (in the case of WNV). If this means that zol-activated Vd2 cells can kill WNV-infected cells, the situation is the same for SARS-coV-1. In the case of ZIKV, phosphoantigen-activated cells are said to have similar activity, thus this could be indicated as well (cf lines 231-232).

Answer 1: we corrected the sentence in the figure.

 Minor corrections

General comment (suggestion):

Question 1: It might be interesting to discuss how the study of these “emerging viruses” adds to the current knowledge of the role of gd T cells in viral infections and how responses in emergent infections differ from responses in “old viruses“ infections ?

Answer 1: According to the reviewer’s suggestion, we added a sentence in the conclusions, line 351-353

Additional spelling or syntax errors (among many others), or sentences benefiting reformulation:

Question 2: L.41 :  circulating BLOOD ?

Answer 2: I’m sorry but we did not find “circulating blood” in the text.

Question 3: L.42: According to ?

Answer 3: We modified the test line 66

Question 4: L62 : “these emerging viruses” : seems to refer to the ones listed at the beginning of introduction and not to the ones just listed before and comprising HIV, HBV, HCV, IAV. This must be clarified.

Answer 4: we clarified the sentence line 71

Question 5: L.72: Virions are spherical and HAVE a large spike.

Answer 5: we corrected accordingly, line 81

Question 6: L.79: there are NO data in THE literature concerning their involvement against MERS-CoV.

Answer 6: we corrected accordingly, line 88

Question 7: L.82:  In September 2003, the SARS-CoV had infected = remove pandemic

Answer 7: we corrected accordingly, line 92

Question 8: L86-89   Keep the transmission route in one sentence and symptoms in a separate sentence

Answer 8: we modified the test accordingly line 97-98

Question 9: L.99: Health care workers WHO experienced SARS-CoV

Answer 9: we modified the test accordingly line 109

Question 10: L.107: On April 8th 2022, 494.587.638 cases are confirmed together with 6.170.283 deaths

Answer 10: we modified the test accordingly line 118-119

Question 11: L.19: As of April 5th 2022

Answer 11: we modified the test accordingly line 120

Question 12: L.110: COVID-19 is A serious public health concern AND demanded

Answer 12: we modified the test accordingly line 121-122

Question 13: L.117: response, remove “and”

Answer 13: We modified the test accordingly

Question 14: L.119: damages

Answer 14: we modified the test accordingly line 130

Question 15: L.130: γδ.T cell count  …?  L.133: 2 weeks after hospital admission

Answer 15: we modified the test accordingly, line 145 and line 147

Question 16: L.135: a high frequency. in comparison to circulating

Answer 16: we corrected the test accordingly, line 150

Question 17: L.138: phenotype

Answer 17:  we corrected the test accordingly, line 152

Question 18: L.153: ref 43 did not look at γδ  distribution/Activation in regard to COVID-19 infection

Answer 18:  we agree with the reviewer, the ref 43 is about the genus Flavivirus in the section of Flavivirus:  43,  Kuno, G.; Chang, G-J.J.; Tsuchiya, K.R.; Karabatsos, N.; Cropp, C.B. Phylogeny of the Genus Flavivirus. J Virol1998, 72, 1:73-83. doi: 10.1128/JVI.72.1.73-83.1998.

Question 19: L.170: of the virus to which ?

Answer 19: we added “When” line 194

Question 20: L.172:  the sentence is either incomplete or improperly phrased.

Answer 20: we modified the test according to the reviewer suggestion, line 195

Question 21: L.185: “Backbone fever” is not appropriate as a symptom.

Answer 21: we modified the test according to the reviewer suggestion, line 212

Question 22: L.190: cytokine storm resulting in ?

Answer 22: we modified the test according to the reviewer suggestion, line 217

Question 23: L.198: VD2 T cells IFN-g production is impaired?

Answer 23: we modified the test according to the reviewer suggestion, line 225

Question 24: L.203: belonging to

Answer 24: we modified the test according to the reviewer suggestion, line 232

Question 25: L.204: Zika forest

Answer 25: we reported the word Ziika in the Luganda languange. We added Zika word, line 233

Question 26: L.212: transfer ? Zika to the baby resulting in…

Answer 26: we modified the test according to the reviewer suggestion, line 241

Question 27: L.214: and studies

Answer 27: we modified the test according to the reviewer suggestion, line 243

Question 28: L.214: in blood of asymptomatic French Polynesia donors

Answer 28: we modified the test according to the reviewer suggestion, line 244

Question 29: L.220: which encode

Answer 29: I’m sorry but we did not find “circulating blood” in the text.

Question 30: L.222: T cells differentiate into Th1

Answer 30: we modified the test according to the reviewer suggestion, line 251

Question 31: L.228: enriched in granzyme B and able to produce

Answer 31: we modified the test according to the reviewer suggestion, line 257-258

Question 32: L.244: humans resulting in

Answer 32: we modified the test according to the reviewer suggestion, line 273

Question 33: L.246: Severe disease includes

Answer 33: we modified the test according to the reviewer suggestion, line 275-276

Question 34: L.249: Do the authors mean that WNV infects the CNS and causes severe disease in immunocompromised or elderly patients? The sentence is not clear.

Answer 34: we modified the test accordingly, line 278-279

Question 35: L.250: A role for γδ  t cells in

Answer 35: we corrected the test accordingly, line 279

Question 36: L.272: with 99% of fatalities occurring in neighbouring …

Answer 36: we corrected the test accordingly, line 301

Question 37: L.288: discarge ???

Answer 37: we corrected the test accordingly, line 317

Question 38: L.290: independently of the clinical out come

Answer 38: we corrected the test accordingly, line 319

Question 39: L.297: belonging to

Answer 39: we modified the test accordingly, line 331

Reviewer 2 Report

With COVID-19 infections being an international issue for more than two years, a review focusing on γδ T cells in viral infections is significant.

Most chapters are very well written and commendable.

If possible, please add any differences in findings between before and after the spread of Omicron strains during COVID-19 infection.

Author Response

To the Editor,

Viruses

Rome, May 20th 2022

Reviewer 2

With COVID-19 infections being an international issue for more than two years, a review focusing on γδ T cells in viral infections is significant.

Most chapters are very well written and commendable.

Question: If possible, please add any differences in findings between before and after the spread of Omicron strains during COVID-19 infection.

Answer: According to the suggestion, we added a sentence in the Section 2.2 line 168

Reviewer 3 Report

The authors have discussed about the pleiotropic role of γδ T cells in emerging viral infection. Specific points that the authors need to address are as follows:

  1. The novelty of the article should be clearly highlighted as number of excellent reviews have already been published on this topic.
  2. More references from last few years should be added to improve visibility and quality of current work.
  3. The search strategy used for the literature review should be indicated.
  4. The pharmacological strategies that can be used to facilitate the use of γδ T cells for immunotherapies should be discussed.
  5. The authors should provide their own justification and relevance of the study. This will help the readers to understand the importance of the paper.
  6. The manuscript should be carefully checked for typographical errors.

Author Response

Reviewer 3

The authors have discussed about the pleiotropic role of γδ T cells in emerging viral infection. Specific points that the authors need to address are as follows:

  1. Question: The novelty of the article should be clearly highlighted as number of excellent reviews have already been published on this topic.

Answer 1. We agree with the reviewer’s suggestion and we added a sentence in the Conclusion Section, line 341-348

  1. Question: More references from last few years should be added to improve visibility and quality of current work.

Answer 2. According to the reviewer’s suggestions, we added new references: n 6-7, 9-10, 14-15, 17, 21-22, 27-29.

  1. Question: The search strategy used for the literature review should be indicated.

Answer 3: We did not specify in the text the searching strategy, as this is not a systematic review. Anyway, the search strategy used for the literature review was: [gammadelta T cells] and/or [viruses] and/ or [emerging viruses]; [gammadelta T cells] and [the single emerging virus].

  1. Question: The pharmacological strategies that can be used to facilitate the use of γδ T cells for immunotherapies should be discussed.

Answer 4. We added a sentence in the Introduction Section, line 66-70 and 354

  1. Question: The authors should provide their own justification and relevance of the study. This will help the readers to understand the importance of the paper.

Answer: we added a sentence in the Conclusion Section, line 341-348

  1. Question: The manuscript should be carefully checked for typographical errors.

Answer: we carefully checked the manuscript for typographical errors.

Round 2
